# Need for a Standardized Translational Drug Development Platform: Lessons Learned from the Repurposing of Drugs for COVID-19

**DOI:** 10.3390/microorganisms10081639

**Published:** 2022-08-12

**Authors:** Frauke Assmus, Jean-Sélim Driouich, Rana Abdelnabi, Laura Vangeel, Franck Touret, Ayorinde Adehin, Palang Chotsiri, Maxime Cochin, Caroline S. Foo, Dirk Jochmans, Seungtaek Kim, Léa Luciani, Grégory Moureau, Soonju Park, Paul-Rémi Pétit, David Shum, Thanaporn Wattanakul, Birgit Weynand, Laurent Fraisse, Jean-Robert Ioset, Charles E. Mowbray, Andrew Owen, Richard M. Hoglund, Joel Tarning, Xavier de Lamballerie, Antoine Nougairède, Johan Neyts, Peter Sjö, Fanny Escudié, Ivan Scandale, Eric Chatelain

**Affiliations:** 1Mahidol Oxford Tropical Medicine Research Unit, Faculty of Tropical Medicine, Mahidol University, Bangkok 10400, Thailand; 2Centre for Tropical Medicine and Global Health, Nuffield Department of Medicine, University of Oxford, Oxford OX3 7LG, UK; 3Unité des Virus Émergents (UVE), Institut de Recherche pour le Développement (IRD), Aix-Marseille University, 190-Inserm 1207, 13005 Marseille, France; 4Laboratory of Virology and Chemotherapy, Department of Microbiology, Immunology and Transplantation, Rega Institute for Medical Research, Katholieke Universiteit Leuven, 3000 Leuven, Belgium; 5Institut Pasteur Korea, 16, Daewangpangyo-ro 712 beon-gil, Bundang-gu, Seongnam-si 13488, Korea; 6Departmet of Imaging and Pathology, Katholieke Universiteit Leuven, Translational Cell and Tissue Research, 3000 Leuven, Belgium; 7Drugs for Neglected Diseases Initiative (DND*i*), 1202 Geneva, Switzerland; 8Centre for Excellence in Long-Acting Therapeutics (CELT), Department of Pharmacology and Therapeutics, University of Liverpool, Liverpool L69 7ZX, UK; 9Global Virus Network (GVN), Baltimore, MD 21201, USA

**Keywords:** COVID-19, drug repurposing, translational medicine, pandemics, clinical trials

## Abstract

In the absence of drugs to treat or prevent COVID-19, drug repurposing can be a valuable strategy. Despite a substantial number of clinical trials, drug repurposing did not deliver on its promise. While success was observed with some repurposed drugs (e.g., remdesivir, dexamethasone, tocilizumab, baricitinib), others failed to show clinical efficacy. One reason is the lack of clear translational processes based on adequate preclinical profiling before clinical evaluation. Combined with limitations of existing in vitro and in vivo models, there is a need for a systematic approach to urgent antiviral drug development in the context of a global pandemic. We implemented a methodology to test repurposed and experimental drugs to generate robust preclinical evidence for further clinical development. This translational drug development platform comprises in vitro, ex vivo, and in vivo models of SARS-CoV-2, along with pharmacokinetic modeling and simulation approaches to evaluate exposure levels in plasma and target organs. Here, we provide examples of identified repurposed antiviral drugs tested within our multidisciplinary collaboration to highlight lessons learned in urgent antiviral drug development during the COVID-19 pandemic. Our data confirm the importance of assessing in vitro and in vivo potency in multiple assays to boost the translatability of pre-clinical data. The value of pharmacokinetic modeling and simulations for compound prioritization is also discussed. We advocate the need for a standardized translational drug development platform for mild-to-moderate COVID-19 to generate preclinical evidence in support of clinical trials. We propose clear prerequisites for progression of drug candidates for repurposing into clinical trials. Further research is needed to gain a deeper understanding of the scope and limitations of the presented translational drug development platform.

## 1. Introduction

The coronavirus Severe Acute Respiratory Syndrome Coronavirus 2 (SARS-CoV-2) is the causative pathogen of COVID-19, one of several coronaviruses that can infect humans, along with SARS-CoV and MERS-CoV [1]. Due to its high infectivity and ability to cause severe disease and death, COVID-19 became a public emergency of international concern and was classified as a pandemic by the World Health Organization (WHO) in March 2020 [2]. At the time of writing, over 537 million confirmed cases and more than 6 million deaths have been recorded globally [3,4], contributing to 14.9 million excess deaths associated with the COVID-19 pandemic [5].

COVID-19 has a polymorphic clinical presentation, ranging from non-specific viral symptoms with fever, cough, dyspnea, vomiting, and fatigue, to isolated upper airway involvement and finally to acute respiratory distress syndrome [6]. Early infection may be initially asymptomatic or evolve in progressive stages (mild and severe). A worsening of symptoms usually occurs 7 to 10 days after initial clinical symptoms, possibly linked to a cytokine storm resulting from a strong immune response and accompanied by a high risk of thrombosis [7]. 

COVID-19 is a serious respiratory disease. Approximately 80% of patients infected remain mildly symptomatic whilst 20% present more severe symptoms [8]. Patients with severe symptoms may require hospitalization, often with oxygen supplementation in critical cases, invasive ventilation, and Intensive Care Unit (ICU) admission. Worldwide, the SARS-CoV-2 pandemic has placed ICUs under immense strain and led to high mortality, until the availability of vaccines to physicians and at-risk patients by the end of 2020 and to the wider population by the first quarter of 2021. In low resource setting areas, where there is a lack of access to such medical tools and infrastructure, the need to prevent progression to severe COVID-19 is critical. Moreover, disruption in available healthcare systems has exacerbated cases of other diseases, such as malaria, human immunodeficiency virus (HIV), and tuberculosis, reinforcing the importance of quickly treating COVID-19 to decrease the spread of the disease. 

DND*i* and several expert groups from the COVID-19 Clinical Research Coalition have harmonized efforts to launch the ANTICOV clinical trial platform in multiple African countries [9]. Initiated in May 2020, ANTICOV focused on identifying and proposing repurposing drug candidates to generate data supporting treatment strategies for mild to moderate SARS-CoV-2 patients in low- and middle-income countries [10]. The ANTICOV clinical trial responded to the urgent need to identify treatments for mild and moderate cases of COVID-19 that can be used early to prevent hospitalization spikes that could overwhelm fragile and overburdened health systems. The ultimate goal was to reduce the number of patients progressing to severe COVID-19 requiring hospitalization, thereby relieving the burden on healthcare systems and facilitating the so-called “flattening of the curve”, considering global public health information available at the time through various open-data-sharing platforms, in particular the coronavirus resource center of the John Hopkins University of Medicine [4]. This objective applies to existing and subsequent waves of the pandemic in contexts where systematic measures such as quarantine or use of personal protective equipment is not feasible. ANTICOV is a flexible and innovative trial designed for treatments to be added or removed as evidence emerges. The trial accommodates testing for multiple antiviral and anti-inflammatory agents and aims to select the most promising treatments from ongoing global scientific efforts with proof of efficacy.

In the traditional (de novo) drug discovery process, it typically takes 10–17 years to bring a drug to the market, and costs reach approximately USD 800 million [11]. In contrast, drug repurposing (also known as repositioning), facilitates the identification of new medications by investigating currently approved drugs for new therapeutic clinical use [11]. This approach is of particular interest in response to the COVID-19 pandemic emergency, where rapid drug discovery for the management of COVID-19 and prevention of disease progression is crucial. Advantages over the traditional drug discovery and development pipeline include lower associated risks owing to previously established pre-clinical, pharmacokinetic (PK), and safety profiles. In addition, since clinical data are available at the beginning of a development project, the risks associated with further development are greatly reduced [12]. Hence, repurposed drugs can be fast-tracked through to Phase 3 human clinical trials [13,14], shortening development time frames and lowering investment costs. 

Historic examples of successfully repurposed drugs include sildenafil (originally indicated for angina, now indicated for erectile dysfunction) or thalidomide (originally indicated for morning sickness, now indicated for multiple myeloma) [15]. Regarding the treatment of COVID-19, the exploration of drugs from the antiviral field provides logical short-term opportunities to rapidly combat an evolving global pandemic. Repurposing existing drugs to treat COVID-19 is biologically reasonable, as SARS-CoV-2 shares some similarities with other coronaviruses, such as SARS-CoV and MERS-CoV [16]. There are successful precedents in repurposing antivirals for new virus targets [17], and indeed, most of the drugs currently in clinical trials for COVID-19 are repurposed approved antiviral drugs. “Obvious” disease-modifying drugs (i.e., anti-inflammatory drugs) such as dexamethasone or baricitinib were successfully repurposed quickly for severe COVID-19 [18,19,20]. Anticoagulant and/or antiplatelet therapies have also been assessed during the pandemic, given that venous thromboembolism and thrombosis complications were observed in COVID-19 patients. While survival increase was shown in non-critically ill hospitalized COVID-19 patients receiving anticoagulant therapy, antithrombotic preventive treatment is not recommended for non-hospitalized patients without evidence of venous thromboembolism [21,22,23]. “Direct repurposing” of existing drugs at the already approved dose was a logical strategy to rapidly identify suitable treatment(s) for the management of severe cases of COVID-19 and to prevent disease progression. Repurposing can also refer to a non-approved dose of an existing drug—“indirect repurposing”—since the approved dose is optimized and selected for another pathogen or indication [24]. 

Historically, the relevance of drug combination regimens has been demonstrated in the fields of HIV chronic viral infection and cancer [25,26]. Exploring drug combinations for COVID-19 also holds much promise, considering the complexity of the disease pathology. Indeed, using drugs with different mechanisms of action (MoA) to target different aspects of the disease (e.g., reducing viral load with an antiviral and suppressing the cytokine storm with an anti-inflammatory compound or immunomodulator) or different steps in the viral cycle would be expected to improve therapeutic efficacy and allow a much wider patient population with different needs to benefit from treatment [27]. Repurposed antiviral and immunomodulator combination regimens have the potential to serve an urgent need in outpatients, both as prophylaxis and as treatment, and is the current focus of the ANTICOV adaptive platform trial in low resource settings [10].

Leveraging existing assets and networks to fast-forward the development of novel COVID-19 treatments, DND*i* designed and implemented a translational drug development platform to generate preclinical evidence in support of treatment arms for the ANTICOV clinical trial. This partnership included well-established virology laboratories and PK modeling expert institutions to systematically evaluate the clinical potential of repurposed drugs. The aim of this COVID-19 translational platform is to catalyze translational and drug repurposing research and accelerate the sharing of data. This includes preclinical data related to in vitro and ex vivo antiviral potency, in vivo efficacy in a hamster infection model, as well as drug metabolism and pharmacokinetic (DMPK) studies of selected repurposed drugs. In addition, the use of pharmacometric modeling and simulation was incorporated for the evaluation of drug exposures in plasma and target organs based on both preclinical and clinical data. 

The current article discusses the processes and methodology we implemented in repurposing drugs in the context of the COVID-19 pandemic and provides examples of identified repurposed antiviral drugs that were tested in our multidisciplinary collaboration with the aim of highlighting areas that need finessing in urgent antiviral drug development.

## 2. Experimental Verification of Repurposed Drugs against COVID-19

As for any drug repurposing development, COVID-19 drug repurposing needed to go through three key stages before progressing to later stages of clinical development: (i) identification of drug candidates, (ii) experimental verification of drug candidates in pre-clinical models, and (iii) assessment of the drug effectiveness in Phase 2 clinical trials [28]. The methodology used in this collaboration to test repurposed and experimental drugs, alone or in combinations, is outlined below (refer to Appendix A). The ideal strategy for the generation of new preclinical data for repurposed drugs against SARS-CoV-2 to build a rationale for clinical evaluation is outlined in Figure 1.

### 2.1. Identification and Selection of Drug Repurposing Candidates for Preclinical Studies

The approach taken by DND*i* to identify and select repurposing candidates for entry into the ANTICOV trial platform was based on publicly available literature (for details see Appendix A). Specific criteria included: (i) potential in vitro activity against SARS-CoV-2; (ii) in vivo efficacy in preclinical species, when available (e.g., antiviral activity of compounds in the Syrian hamster or other appropriate animal models); (iii) their safety in another indication (effects of common comedications or comorbidities as well as risks associated with pregnancy or lactation, and compatibility with anti-inflammatory combination medicines were considered); (iv) route of administration; (v) their ability to be manufactured at scale and at an affordable cost; and (vi) in the best case scenario, preliminary evidence of their activity in mild to moderate SARS-CoV-2 patients from proof-of-concept (PoC) studies, antiviral studies, and modeling data supporting the best dose regimen.

A preliminary longlist of candidates included drugs demonstrated to have in vitro antiviral activity against SARS-CoV-2 virus and for which the plasma concentration following approved dosing in humans exceeded that needed for antiviral activity in vitro. For antiviral drug candidates, free plasma exposures at the highest dose approved by regulatory agencies (mainly the US Food and Drug Administration [FDA] and the European Medicines Agency [EMA]) were used to estimate the potential to achieve free human plasma coverage above the SARS-CoV-2 antiviral EC_50_. This provided an initial prioritization of the longlist by excluding/down-selecting compounds with a low probability of achieving clinical antiviral efficacy. Drug protein binding was assumed to be minimal in the in vitro system and no correction of the in vitro EC_50_ efficacy was made at this stage; the free drug hypothesis did not always apply, and data should be interpretated with caution and in a drug-specific manner [29]. For the final stage selection, sustained exposure over the in vitro antiviral EC_90_ was targeted. The lung was a key target organ of the SARS-CoV-2 virus during the early stages of the pandemic; therefore, the initial prioritization based on free plasma levels was followed by modeling of local exposure in the lung as described in previous reports [30,31]. Of note, compounds suitable for treatment of late stage COVID-19 were not included in this analysis (e.g., dexamethasone and baricitinib).

During the progress of this work, additional compounds were continuously evaluated and considered for inclusion as new data were made available from preclinical and clinical investigations. This was a highly dynamic period of research during 2020–2021, which witnessed an explosion of activity and rapid sharing of results, often through preprints and other non-peer reviewed releases with higher levels of uncertainty and a greater need for independent corroboration.

The selection of drug repurposing combinations was based on the hypothesis that a combination of a SARS-CoV-2 antiviral compound and an anti-inflammatory agent able to deescalate the inflammatory response caused by the viral infection would maximize the treatment impact and prevent disease progression.

### 2.2. Generation of Preclinical Data

Following the selection of candidates described above, the following drugs were sourced from various providers: atazanavir, favipiravir, sofosbuvir, ritonavir, nitazoxanide, ambroxol HCl, camostat mesylate, cepharantine, and fluvoxamine maleate from Ambeed Inc., USA; ritonavir, nitazoxanide, atazanavir, N-desethyl-amodiaquine, and nelfinavir from BLD Pharmatech Ltd.; daclatasvir, sofosbuvir and its metabolite, nitazoxanide metabolite tizoxanide, pentoxyfilline, and clofazimine from AK Scientific, Union City, CA, USA; nelfinavir mesylate, molnupiravir and its main metabolite from MedChemExpress, Monmouth Junction, NJ, USA; colchicine, amodiaquine, mefloquine HCl, and ivermectin from Carbosynth Ltd., Compton, UK; bemnifosbuvir (free base AT-511, salt AT-527, and its metabolite AT-273) from DC Chemicals, Shanghai, China. Nirmatrelvir was synthesized at TCG Lifesciences Private Ltd., Kolkata, India. Preclinical data were then generated, including pharmacodynamic (PD) in vitro assays, ex vivo 3D models, and in vivo hamster disease models, to test potency against SARS-CoV-2 [32]. Pharmacokinetic data were also generated, including drug exposure in plasma and tissue of interest (lungs). Finally, population PK modeling and simulation was carried out, as described below.

#### 2.2.1. In Vitro/Ex Vivo Activity

In vitro profiling of the selected repurposed and experimental drugs was performed in three different phenotypic virus-cell-based antiviral assays with SARS-CoV-2 in Vero cells (+/− CP-100356, a Pgp-pump inhibitor), Calu-3 cells using SARS-CoV-2/KCDC03 (Wuhan strain), and A549-Dual™ hACE2-TMPRSS2 cells using SARS-CoV-2-B.1.1.7 (Alpha variant). Methods and protocols were rapidly adapted from previous antiviral screening campaigns (MERS, SARS-CoV) for developing approaches to identify antivirals against SARS-CoV-2. These assays were designed to determine the effect of small molecules on the infection and replication of SARS-CoV-2 and relied either on the assessment of the cytopathic effect (CPE) of the virus or on quantification of the viral N protein (immunofluorescence-based assay) in the presence of different drug concentrations. Experimental details are provided in the Appendix A.

Ex vivo 3D modeling was carried out using Human Airway Epithelia (HAE) in human Air Liquid Interface (ALI) cultures (Mucil AIR—Epithelix™) infected with SARS-CoV-2 ancestral strain B.1 (Bavpat1) or B.1.1.7 (Alpha variant). These differentiated ALI models were designed to study respiratory virus pathogenesis and to evaluate antiviral toxicity and efficacy. The model holds in vitro specific mechanisms to counter invaders comparable to the in vivo situation, such as mucus production, mucociliary clearance, and secretion of defensive molecules [33]. Experimental details are provided in the Appendix A.

#### 2.2.2. In Vivo Efficacy in Hamsters

In vivo efficacy studies for all tested drugs were conducted in hamsters. Features associated with SARS-CoV-2 infection in hamsters recapitulate some characteristics found in humans with mild SARS-CoV-2 infections; the hamster model was therefore considered an adequate animal model for these studies (hamster pathology description following infection and experimental details are provided in Appendix A). Modeling of the relevant clinical dosing regimen was selected based on the labels of the considered drugs, assuming 10 days of dosing in line with the target product profile [34]. The hamster infection model of SARS-CoV-2 was used as previously described [35,36,37] using either the Wuhan strain, the ancestral SARS-CoV-2 B.1 (Bavpat1), or SARS-CoV-2-B.1.351 (Beta variant) strains. Drugs were administered to infected hamsters as homogenous suspensions in their adequate respective vehicle. Groups of 6 hamsters were infected intranasally with 10^4^ TCID_50_ of SARS-CoV-2. Groups of 6 animals received oral doses of repurposed drugs based on body weight. The negative control groups of 6 hamsters were treated with the respective vehicle. Lastly, the positive control groups of 6 animals were treated with favipiravir intraperitoneally (926 mg/kg/day BID), based on doses identified in a hamster model by Driouich et al. [38]. Following infection, hamsters were treated for 3 days, on day 0, day 1, and day 2 post-infection (dpi). Infectious titers were measured using a TCID_50_ assay, and the viral RNA yields were measured by quantitative real-time PCR in clarified lung homogenates or plasma.

The impact of repurposed drugs on lung pathological changes induced by SARS-CoV-2 was also independently explored. Groups of 4 hamsters were treated orally with the selected repurposed drug. Untreated groups of 4 hamsters received the corresponding vehicle BID. Animals were sacrificed at 5 dpi and based on severity of lung inflammation, alveolar hemorrhagic necrosis, and vessel lesions, a cumulative score from 0 to 10 was calculated and assigned to a severity grading (0 = normal; 1 = mild; 2 = moderate; 3 = marked; 4 = severe; refer to experimental details provided in Appendix A.

#### 2.2.3. Protein Binding and Correction of IC_50_ Values for Protein Binding

Plasma protein binding of repurposed drugs in hamsters and humans was either measured by the Rapid Equilibrium Dialysis (RED) as described in Abdelnabi et al. [35] or was available from the literature. In addition, binding to assay medium (2% serum DMEM supplemented with 2% *v*/*v* FCS) in antiviral assays (A549-ACE2TMPRSS2 cells) was determined by RED. In vitro IC_50_ values were first corrected for protein binding in assay media according to: IC_50unbound_ = IC_50 media_ × f_unbound, media,_
(1)
where IC_50unbound_ is the free drug concentration needed to achieve 50% inhibition of the virus-reduced enhanced green fluorescent protein (eGFP) signals compared with the untreated virus-infected control cells, and f_unbound, media_ is the unbound drug fraction in assay media.

In a second step, values were scaled to represent the total plasma concentrations in humans or hamsters needed to achieve 50% of the maximum effect, according to: IC_50 plasma_ = IC_50 unbound_/f_unbound, plasma,_(2)
where f_unbound, plasma_ is the unbound drug fraction in plasma.

#### 2.2.4. Pharmacokinetic Studies in Hamster

Pharmacokinetics of selected drugs were assessed in uninfected female LGV Golden Syrian hamsters (satellite PK study) purchased from Beijing Vital River Laboratory Animal Technology Co., Ltd. (Beijing, China) or Envigo (Indianapolis, Indiana, USA). Details regarding dosing regimen, drug formulation, PK sampling time points, as well as bioanalytical procedures are provided in the Appendix A). Briefly, most drugs were given as a single dose by oral gavage except for nelfinavir (intraperitoneally), amodiaquine, and ivermectin (both oral and subcutaneous administration). For nelfinavir and atazanavir (ritonavir-boosted), single and multiple dose PK studies were performed. Three dose levels (triplicate experiments in each dose level) were tested for the majority of drugs, with doses ranging from 1 to 200 mg/kg, depending on the drug. For nelfinavir and ivermectin, PK profiles at two dose levels were assessed. Blood samples (1–7 samples per hamster, depending on the drug) were collected and drug concentrations analyzed with a LC–MS/MS method, which was optimized for the various drugs.

#### 2.2.5. Population Pharmacokinetic Analysis

For each drug, plasma concentration–time profiles from satellite PK studies in hamsters were pooled and analyzed using a nonlinear mixed-effects modeling approach in NONMEM, v7.4 (Icon Development Solution, Ellicott City, MD, USA). Details regarding model development and validation are provided in the Appendix A. Briefly, different absorption, disposition, and variability models were evaluated. To improve the translational aspect of the model, body weight was implemented a priori as an allometric function. Moreover, dose and (if applicable) different routes of administration were evaluated as covariates. Parent and metabolite data (if available) were analyzed simultaneously, assuming complete in vivo conversion. 

In addition to the population PK analysis in hamsters, a literature search was performed to identify available PK information in humans [39]. Details regarding the model selection (if a population PK model was published) or model derivation (based on non-compartmental PK analysis or digitized PK data) are provided in the Appendix A.

#### 2.2.6. Population Pharmacokinetic Simulations

PK parameters from the final population PK models in humans and hamsters were used to simulate plasma exposures for various dosing scenarios. Simulations were performed with the mlxR package in the R software [40], for a treatment duration of 10 days in both hamsters and humans. A total of 1000 hypothetical individuals (body weight of 70 kg) or hamsters (body weight 120 g) were simulated for each dosing scenario.

For the simulations in humans, inter-individual variability (IIV) as reported in clinical PK models was implemented (when available). Otherwise, an IIV of 15% was assumed for all PK parameters (in humans and hamsters) to reflect an expected population variability. Simulations of plasma exposure in humans were based on the standard (clinical) dose given once or twice daily according to the general practice. A loading dose was not considered. 

Dosing scenarios for simulations in hamsters correspond to the dose range tested in the satellite PK studies in hamster; and twice daily dosing was assumed (Appendix A for pharmacokinetic profiles in hamsters). When required, extrapolation beyond this dose range was performed to identify doses in hamsters that reach drug exposure in humans at clinically relevant doses. Peak concentrations (C_max_) and cumulative areas under the concentration-time curves (AUC_total_) were extracted and used for comparison with clinical simulations. Boxplots of simulated C_max_ in humans and hamsters were overlaid with the in vitro IC_50_ values reported in A549-ACE2TMPRSS2 cells. These IC_50_ values were corrected for protein binding (see above) to compare simulated C_max_ from the developed PK models with expected therapeutic concentrations.

## 3. Results

### 3.1. Selected Repurposed Drugs for Further Evaluation

The initial longlist of antiviral drug candidates identified 88 compounds for further investigation. Following the prioritization process using criteria listed in 2.1 (Identification and selection of drug repurposing candidates for preclinical studies), 25 compounds were selected as potential candidates for inclusion in the ANTICOV study (Appendix A). Remdesivir was not considered further because its administration mode by infusion in humans was not considered adequate for inclusion in ANTICOV; moreover, it was not deemed a suitable control for Syrian hamster experiments due to difficulty of administration and lack of efficacy in standard rodent animal models of SARS-CoV-2 infection related to poor plasma stability in this species. The remaining compounds were further evaluated in the preclinical investigations summarized in Table 1. Selected repurposed drugs were grouped according to compound class for single agents (Table 1), namely direct-acting antivirals (DAAs) and indirect-acting antivirals (IAA). The selected drug candidates covered a wide variety of MoAs. DAA compounds included HIV protease inhibitors, RNA viral polymerase inhibitors, hepatitis C virus (HCV) NS5A inhibitors, protease (Mpro) inhibitors, and nucleotide polymerase inhibitors. IAA compounds comprised mucolytic, antimalarial, ACE binding, anti-inflammatory, anti-parasitic, and vasodilatory drugs as well as selective serotonin reuptake (SSRI) inhibitors or combinations thereof (Table 2).

Based on the outcome of an expert panel discussion involving representatives from various organisations (Access to COVID-19 Tools Accelerator [ACT-A] Therapeutics Partnership, Unitaid, Wellcome on behalf of the ACT-A), the following drug combinations were initially prioritized: atazanavir/ritonavir (ATZ/r) + nitazoxanide; favipiravir + ATZ/r; favipiravir + nitazoxanide; and sofosbuvir/daclatasvir. During the project, we continuously reviewed new options and promising additional re-purposed drug candidates or drug combinations based on newly generated preclinical and proof-of-concept clinical data. Clinical results highlighted a potential for inhaled steroids, budesonide and ciclesonide, in COVID-19; these were selected as the primary anti-inflammatory combination partners for the ANTICOV trials [41,42,43]. However, inhaled steroids were not assessed in our translational platform, as drug delivery by inhalation was not deemed suitable for assessment in animal models. Ongoing review highlighted the rationale and potential for the following combinations, which were further assessed preclinically: amodiaquine + ivermectin following data showing potential for a combination [44]; nelfinavir + cepharantine, selected following published in vitro data highlighting a potential synergistic effect of this combination in limiting SARS-CoV-2 proliferation [45] and the impact of nelfinavir on lung inflammation in animal models, although it had no impact on SARS-CoV-2 viral load [46]; and clofazimine, selected for combination with the experimental drug molnupiravir, because it showed anti-SARS-CoV-2 potency in vitro in Vero cells and the property to accumulate in the lung [47,48].

### 3.2. Preclinical Data Generated for DAAs

The in vitro antiviral activity of all selected drugs was assessed using three different cell lines, namely Vero kidney epithelial cells (extracted from an African green monkey), Calu-3 cells (extracted from male hepatoma tissue), and A549 cells (adenocarcinoma human alveolar basal epithelial cells). Results are depicted in Table 1, and more details including standard deviations are provided in Appendix A.

Regarding the selected DAA compounds, only molnupiravir (RNA-dependent RNA polymerase (RdRp) inhibitor) and its metabolite EIDD-1931 as well as nirmatrelvir (SARS-CoV-2 main protease—Mpro also known as 3CLpro-inhibitor) showed high potency (in the following defined as EC_50_ or IC_50_ < 10µM) in all three in vitro assays without limitations of toxicity. Notably, molnupiravir potency (IC_50_ < 10 uM) was only reached in the presence of P-gp efflux inhibition. With respect to nelfinavir, high potency in Vero cells and the A549 cell was associated with some toxicity, complicating the distinction between the antiviral effect versus overall toxicity (Table 1). Similarly, daclatasvir showed high potency associated with toxicity in Vero cells, while no potency was observed in the other cell lines. None of the other DAA compounds (atazanavir, bemnifosbuvir, favipiravir, and sofosbuvir) demonstrated potency in the investigated cell lines (Table 1).

The ranking observed for the DAA compounds based on the in vitro cellular assays was consistent with the screening assessment in primary human airway epithelial cells (HAE): molnupiravir (10 µM), nirmatrelvir (1 µM), and nelfinavir (10 uM) were the only DAA compounds that completely inhibited viral replication for the entire duration of the experiment (4 days) when added to the culture medium at the basolateral site of the ALI 1 h before infection (Appendix A).

To verify whether human exposure under standard clinical regimens reached target concentrations, population PK modeling was applied. Exposure targets (C_max_ above IC_50_ under clinical dosing regimen) were set to in vitro IC_50_ values reported in A549-ACE2TMPRSS2 cells, which were corrected for protein binding (Appendix A). In addition to modeling and simulation of human exposure, population PK models based on hamster PK studies were developed for comparative evaluation of plasma exposures. Plasma concentration—time profiles in hamsters used for the development of population PK models for each drug are shown in the Appendix A. Examples of structural models, including model parameterizations, are provided in the Appendix A. All structural models and covariate effects for both humans and hamsters are summarized in the Appendix A, along with the respective population PK parameters (Appendix A). Simulations with the final population PK model were used to specify the dose range in hamsters matching human exposure in plasma (C_max_ and AUC_total_) under a standard clinical dosing regimen (Appendix A). An example of simulated plasma concentration—time profiles in both humans and hamsters is shown in Figure 2 (molnupiravir metabolite EIDD-1931); the corresponding boxplots with simulated C_max_ (overlaid with exposure targets) and AUC_total_ in humans and hamsters are shown in Figure 3. The wider dose range in Figure 3 illustrates the screening process to identify doses in hamsters matching drug exposure in humans at a clinically relevant dose. Boxplots with secondary PK parameters for all other investigated compounds are provided in the Appendix A.

Regarding drug exposure in humans at a clinically relevant dose, only molnupiravir showed simulated median C_max_ values above the IC_50_ corrected for protein binding. This was not assessed for nirmatrelvir because clinical data were not available at the time. In hamsters, favipiravir also reached exposure targets (at a dose matching human exposure). It is interesting to note that for those compounds for which C_max_ exceed exposure targets in either humans or hamsters, clinical efficacy was also observed.

For in vivo efficacy studies in hamsters, higher doses compared to doses matching human exposure were used, except for nelfinavir and sofosbuvir. For example, a dose-escalation study for molnupiravir with up to 200 mg/kg BID for 4 days was performed to test efficacy. However, only a few DAA compounds demonstrated in vivo efficacy in hamsters despite testing higher doses, including favipiravir, molnupiravir, and nirmatrelvir (Appendix A Appendix A). These compounds also reached target exposures (C_max_ above the IC_50_) in hamsters at efficacious dosing regimens (Table 1). Conversely, compounds failing to reach target exposures in hamsters in the PD experiments also showed no activity in the hamster efficacy model (daclatasvir, atazanavir, and sofosbuvir). Extended exposure with trough concentrations (C_min_) above IC_50_ for more than 24 h was only observed with nirmatrelvir. For nelfinavir, no effect on viral load was detected, but a diminution of lung inflammation and “disease outcome” was nevertheless observed.

Although systematic PK/PD analyses were not performed at this stage, some trends regarding the PD driver linking preclinical and clinical activity could be observed, whereby the following parameters were evaluated: exposure in hamsters above IC_50_ for 24 h, activity in the SARS-CoV-2 hamster model, and C_max_ above IC_50_ in a hamster PD experiment (Table 1). It was not possible to conduct comprehensive PK/PD analyses because additional efficacy data in hamsters would have been needed. Efficacious dosing regimens significantly reducing the lung TCID_50_ in comparison to untreated control animals are illustrated in the Appendix A. Data generated here could not be used to derive the likely PD driver to achieve efficacy in the hamsters even though time above a minimum inhibitory concentration, total exposure, or C_max_ are all important parameters. A parallel with humans could be made, as when C_max_ exceeds the IC_50_, clinical efficacy was also observed (evident for molnupiravir but not favipiravir) [49,50,51]. Each of these three PK/PD parameters (exposure in hamsters above IC_50_ for 24 h, activity in the SARS-CoV-2 hamster model, and C_max_ above IC_50_ in the hamster PD experiment) was achieved only for nirmatrelvir. Regarding daclatasvir, ritonavir-boosted atazanavir, and sofosbuvir, none of the exposure targets was reached, and no activity was observed in hamsters.

### 3.3. Preclinical Data Generated for IAAs

From the IAA compounds selected for profiling, only clofazimine, colchicine, and ivermectin demonstrated some potency in all three in vitro cellular assays. However, potency of these compounds was associated with cell toxicity or a very low selectivity index (SI). Amodiaquine, cepharanthine, fluoxetine, and mefloquine (in addition to the abovementioned compounds) showed activity in the Vero cell line but no activity in Calu-3 or A549 cells (Table 1). Moreover, high potency (<10 µM) in Vero cells was associated with toxicity for most compounds with the exception of nitazoxanide. Nitazoxanide also showed high potency and no toxicity in Calu-3 cells, similar to camostat. All remaining IAA compounds (ambroxol, fluvoxamine, pentoxyfilline, probenecid, and proxalutamide) showed no potency in all three investigated cell lines (Table 1 and Appendix A).

Regarding the ex vivo 3D model, only camostat mesylate and nitazoxanide showed activity in the primary HAE cells (10 µM and 5 µM, respectively) (Appendix A).

With respect to drug exposure in humans at a clinically relevant dose, none of the IAA compounds reached exposure targets (Table 1 and Appendix A). Likewise, C_max_ for the dosing regimens in hamsters matching human exposure failed to reach in vitro IC_50_ values corrected for protein binding. At doses tested in hamster efficacy experiments, simulated plasma exposures were below target concentrations. Clofazimine was the only exception (simulated C_max_ was above IC_50_ at 25 mg/kg). None of the IAA compounds demonstrated efficacy in in vivo hamster models (Table 1).

### 3.4. Preclinical Data Generated for Drug Combinations

As discussed, the project involved ongoing review of promising drug combinations based on newly generated preclinical and proof-of-concept clinical data. However, none of the combinations tested in vivo in the SARS-CoV-2 infection hamster model showed additive or synergistic efficacy (Table 2) above that reported for single compounds already shown to be efficacious in this model. The combination of favipiravir with nitazoxanide or atazanavir did not show any additive or synergistic activity compared with favipiravir alone. Similarly, the combination of clofazimine with molnupiravir did not show any additive or synergistic activity compared with molnupiravir alone. Interestingly, combining the two DAAs molnupiravir and nirmatrelvir—currently the only two drugs having received emergency use authorization from the FDA—did not lead to additive or synergistic efficacy at the doses tested (preliminary data).

## 4. Discussion and Lessons Learned

### 4.1. Selection of Repurposed Drugs

The selected repurposed drugs were chosen based on existing preclinical data and included, among others, known antivirals, antimalarials, antiparasitics, and host modulators (Appendix A). While the initial selection of repurposed drugs for COVID-19 has concentrated on registered drugs originally developed for treatment against other viruses (e.g., HCV, HIV) and acting against known targets of the virus cycle (e.g., inhibitors of nsp13 helicase, inhibitors of RNA-dependent RNA polymerase [RdRp]), this did not necessarily prove to be a criterion for preclinical and in some cases clinical activity against SARS-CoV-2 (e.g., atazanavir, sofosbuvir). Indeed, the target sequence might differ or be less relevant due to differences between virus families [52].

Similarly, selection based on in vitro potency (possibly with an inadequate assay in Vero cells) and exposure in the relevant organs was also no guarantee of in vivo efficacy (e.g., clofazimine, atazanavir). In addition, drug exposure measurements observed for standard regimens in humans are not always achieved in the animal model for virus-induced disease, making it complicated to postulate any potential efficacy or to assign a rationale for these drugs from a preclinical point of view. Indeed, hamsters tended to have much higher clearance rates for the drug than that seen in humans, and more frequent dosing than BID is extremely difficult under containment level 3/biosafety level 3 (CL-3/BSL-3) conditions.

### 4.2. Relevance of In Vitro Assays

Over the course of our study, and throughout the COVID-19 pandemic, a lack of standardized assays specially dedicated for SARS-CoV-2 was apparent, as has been reported by other laboratories [53,54]. In vitro assays do not always reflect the potential metabolism or required activation mechanism of the drug for it to show efficacy, hence the importance of knowing and understanding the drug’s underlying mechanism of action.

Immortalized cell lines such as Vero cells offer a readily available system for initial screens. However, as steps of the viral replication cycle might be fundamentally altered in these cell lines, the risk of false positives (such as chloroquine [55]) and false negatives (such as camostat mesylate [56]) is high with in vitro assays. Meaningful counter screens must be integrated into any discovery campaign to validate the physiological relevance of results, for instance by using disease-relevant primary cells or organoid models [57], even though one cannot rule out that other cell systems could also result in false positives or false negatives.

Published in vitro potency data in cell systems have shown that atazanavir (ATZ) and the ATZ/ritonavir combination are active against the SARS-CoV-2 virus [58]. Consistent EC_50_ values of 0.5–9.4 µM and 0.6 µM have been reported in SARS-CoV-2-infected Vero cells and infected A549 lung epithelia cells, respectively [58,59]. However, the choice of host cell used for testing in vitro potency of selected repurposed compounds may have an impact on the assay itself. One of the major limitations of Vero cells is that they are defective in the expression of the main SARS-CoV-2 receptors (angiotensin-converting enzyme 2/ACE2 and transmembrane protease serine 2/TMPRSS2) known to play a pivotal role in SARS-CoV-2 binding and cell entry [60]. In addition, Vero cells express the functionally active P-glycoprotein (Pgp) efflux pump, which could be a source of bias for in vitro studies of drugs with Pgp-mediated drug transport [61]. Moreover, phospholipidosis in Vero cells has been shown for ex vivo cationic amphiphilic drug to correlate with an absence of in vivo success so far, suggesting that relying on this sole MoA would translate into a lack of clinical efficacy of most drugs repurposed to date for SARS-CoV-2. Hence, distinguishing compounds with this confounding effect could accelerate the identification of genuinely potent antivirals against SARS-CoV-2 and other viruses [62].

Human airway epithelial Calu-3 cells were used to understand the pathophysiology of the virus, as they also express the TMPRSS2 receptors. They were shown to be permissive to many coronaviruses, including SARS-CoV-2, and were described as a suitable host cell line that allows the virus to grow in vitro despite a low throughput. Rapidly, the use of lung epithelial cells showed differences in drug sensitivities compared to Vero cells (e.g., hydroxychloroquine versus remdesivir, as described by Dittmar and colleagues [63]). Further studies have demonstrated that ACE2 and TMPRSS2 receptors were expressed in lungs and in bronchial transient secretory cells [64], strongly supporting the importance of using tissue-relevant cell lines to profile any drugs in vitro against the SARS-CoV-2 virus.

Possibly the most relevant and predictive cell line for in vitro potency assays was the A549-TMPRSS2 adenocarcinoma human alveolar basal epithelial cell line, whose receptors are closely related to respiratory cells; SARS-CoV-2 uses the SARS-CoV receptor ACE2 for host cell entry [64,65]. In our experience, A549 airway epithelial cells were the most reliable for in vitro/in vivo translation purposes, with the exception of favipiravir, which did not show any in vitro activity. This drug was, however, used as calibrator for in vivo models as it was the only efficacious drug in the PD models available at the beginning of the pandemic. The example of favipiravir highlights a major limitation observed in the translation of in vitro to in vivo data when it comes to compounds having a complex in vivo metabolism. Similarly, we were unable to reproduce potency data for the bemnifosbuvir (AT-527) experimental compound, which was tested in a Phase 2 trial but failed to meet the primary end point of reduction from baseline in the amount of SARS-CoV-2 virus in patients with mild or moderate COVID-19 compared to placebo [66].

Another ex vivo/in vitro cell model of the human airway epithelium cultured at the air–liquid interface emerged during the pandemic and was widely used. The use of this 3D model demonstrated good prediction of positive efficacy outcome in the PD hamster model with the same exception as above. Favipiravir did not show any ex vivo activity, while being efficacious in the in vivo hamster model; conversely, nelfinavir and nitazoxanide only demonstrated ex vivo potency at high concentration but no in vivo efficacy.

Whenever possible, we highly recommend that drugs be assessed on multiple in vitro potency assays to avoid any confounding factors and to add confidence to the translatability of in vitro data. A drug that demonstrates the same range of IC_50_ on multiple cell lines has a higher potential to show in vivo efficacy if adequate exposure can be reached. Confirmation of in vitro data in several laboratories using the same assay will also build confidence in moving a drug to the next stage, be it in vivo animal models or clinical PoC.

The current analysis reveals that many potential drugs are unlikely to achieve the human target concentrations necessary to adequately suppress SARS-CoV-2 under normal dosing conditions. The data emerging from global screening efforts were not routinely benchmarked and prioritized against achievable concentrations after administration of doses proven to have acceptable safety profiles in humans [30].

Of the many treatment options tested preclinically and clinically, the majority of reports of in vitro activity focused on EC_50_ and did not consider EC_90/95_ or protein-adjusted EC_90/95_ as per antiviral convention [29,30], nor did they consider the achievable concentrations in plasma or relevant compartments such as lungs for COVID-19. EC_50_ alone in plasma may not be a strong enough indicator of antiviral activity. It is critical that candidate medicines emerging from in vitro antiviral screening programs are considered in the context of their expected exposure in humans where possible. Lung accumulation in addition to plasma exposure could provide additional insights regarding therapeutic advantage and reassurance to move forward to clinical trials [30]. However, given the different tropism in newly emerging variants of SARS-CoV-2 such as the Omicron variant sub-lineages, the lung is becoming less relevant as a specific target organ in which antiviral exposure should exceed EC_90_ of the drug.

### 4.3. Relevance of In Vivo Assays

Among the limitations of the repurposing approach is the formulation of the active pharmaceutical ingredient (API) for in vivo study in animal models, since these are often not available in formulations or possible volumes of administration that would exert an adequate exposure in hamster disease models [67]. Moreover, the mode of administration of the drug such as infusion or inhalation can make its assessment especially difficult in BSL-3 laboratories.

Current animal models are limited to the detection of direct antiviral activity. As such, the host-directed effect is not captured. However, one could argue that in some instances, while looking at other readouts, in particular inflammation using histology methods or measuring cytokine release, some positive effects related to disease outcome can be detected, with their relevance yet to be confirmed clinically. Nelfinavir, for example, provided a measure of clinical efficacy by reducing lung inflammation in the hamster model of infection, although no effect on viral load was observed [46]. Our experience showed that there is a need for a clear definition of the targeted stage of the disease, as antivirals are likely indicated in the early disease stage to reduce the viral load, whereas anti-inflammatory drugs are required in late-stage disease to address the cytokine storm or other symptoms emerging from COVID-19 disease. As there is no adequate model to assess anti-inflammatory symptoms and consequences of viral infection, the repurposing approach described focused mainly on the antiviral efficacy of compounds, which represents only part of the approach needed to combat COVID-19. Indeed, depending on patient needs and risks, management of COVID-19 can involve intensive care, oxygen therapy, anticoagulation therapy, steroids, antivirals, or immunomodulation drugs [68]. From a public health perspective, appropriately selected and tested repurposed antiviral drugs can relieve the burden of hospitalizations, but they are not the only available tools against COVID-19.

Our study suggested that it would be beneficial for SARS-CoV-2 activity data to be performed with standardized protocols and with standardized effective concentration (EC) activity values as a marker of the concentrations required to suppress the virus to therapeutically relevant levels [30]. Our investigations highlighted a lack of preclinical data on antiviral efficacy in relevant in vivo models compared to relevant benchmarks (e.g., remdesivir, favipiravir) [38,69]. Drugs for which IC_50_ is not reached by C_max_ or C_min_ also showed no activity in the PD hamster study. For those compounds that reached target exposures, activity in hamsters needed to be confirmed, i.e., simulation can detect true negative but not the true positive (Table 3).

Population PK simulations help suggest relevant preclinical dosing regimens and understand the dose–response relationship. When added to available information regarding safety, availability, and cost, these data played a critical role in the overall assessment of the suitability of possible repurposed treatments. For several compounds (daclatasvir, colchicine, fluvoxamine maleate, mefloquine, and nitazoxanide), our experiments showed that it was not possible to achieve a plasma concentration exceeding a relevant in vitro EC_50_ against SARS-CoV-2 for 24 h, taking into consideration plasma protein binding. This criterion was, however, achieved with nirmatrelvir, which showed a significant reduction of the lung TCID_50_ of SARS-CoV-2 in the hamster model. It is important to note that for the simulations performed, we assumed that the (unbound) free drug hypothesis holds [29].

Interestingly, population PK modeling and simulations performed for bemnifosbuvir (AT-527) experimental drug showed that the IC_50_ reported in the literature [70] was not reached in hamsters at any of the doses tested. This result correlates with a lack of efficacy since no activity in the PD hamster study could be observed.

Another important aspect concerning COVID-19 is the management of patients developing severe or critical disease. For those patients, antiviral therapies are insufficient and must be complemented with corticosteroids, immunomodulators, and supplemental oxygen. The preclinical models described above are not suited to identify such treatments or to make therapeutic recommendations based on disease severity for the management of hospitalized COVID-19 patients.

### 4.4. Working under Time Pressure

Over the course of the COVID-19 pandemic, the scientific community has been called to rise to the occasion with unprecedented intensity and urgency [71]. While growing immunity and the availability of vaccines and first generation DAAs in high income countries has changed the current COVID landscape (even though new virus variants with additional capacity to escape the immune response continue to emerge), we can begin to reflect on the triumphs and challenges related to drug repurposing. In addition to a number of clinical trials not being initially well-suited to a public health emergency, some trials initiated in the early days of the pandemic were underpowered and poorly designed, leading to wasted resources generating false leads as seen in the enthusiastic but ultimately refuted reaction to hydroxychloroquine or ivermectin [72]. Because of the urgency to identify clinical candidates, decisions were made using data generated on existing and available assays (such as in vitro activity assays in Vero cells, as explained above). Additional examples of false leads include sofosbuvir/daclatasvir, lopinavir/ritonavir, and chloroquine that all showed negative results or no effect in clinical trials [73,74,75], while questionable validity of preclinical evidence for atazanavir [58] and amodiaquine [76] still needs robust clinical assessment before completely discarding these two drugs.

The challenges of fast developing treatment strategies were unprecedented during COVID-19. During the pandemic, research in the field was churned out at an unprecedented speed and was complicated by a steady appearance of new variants of concern [77]. Indeed, the SARS-CoV-2 variants used in vivo do not show the same cellular tropism as the current ones (e.g., Omicron sub-lineages). A key issue is that the variants have been emerging faster than the time required for producing data for relevant drugs. Similarly, from a logistical point of view, there is no way of fast-tracking animal models. It takes time to develop specifically dedicated models both in vitro and in vivo. Researchers were confronted with balancing urgency of response versus rational scientific decision while integrating growing knowledge of the disease.

In an effort to be collaborative and transparent, a wealth of data was shared on open access platforms, which was an invaluable contribution to the scientific community. In some instances, again because of the speed of reaction, this fair approach also included making preliminary data (not peer reviewed) accessible that did not always justify the further development of identified drugs. Other clear limitations shown during our study and highlighted above remain the identification of potential drugs of interest having an MoA other than antiviral activity. Achieving clinically relevant exposure in animal models was an additional challenge for the identification of adequate repurposed drugs for the treatment of COVID-19, related among others to the need for optimal formulation and different metabolism in animal species.

## 5. Conclusions

Despite remarkable collaborative efforts, the scientific community lacks standardized preclinical procedures and data packages that can help inform drug choice. We noted that preclinical information at times provided conflicting evidence compared with antiviral activity described in the literature, emphasizing the need for a standardized preclinical data package to better inform evidence-based decision making and later clinical development. For example, a partnership has identified a gap in existing clinical trial pipelines that merits investment in further evidence generation for repurposed antivirals and combinations regimens [78]. This highlights the challenges involved in rapidly identifying antiviral treatments and the need for an improved and standardized drug development process. A unified standardized strategy is necessary for selecting, testing, and validating candidate drugs.

Repurposing drugs for COVID-19 has been limited by an empirical choice based on limited data and often following sparse data from in vitro assays without a strong rationale. As stated by Grobler et al. [53], there is a need for a coordinated systematic approach to identify and test promising drugs. Since repurposed drugs were not optimized for the treatment of COVID-19, clearly defined and harmonized criteria from preclinical data must be implemented to be better prepared and to respond faster in the event of a new pandemic. The necessary short-term dependence on repurposing existing drugs cannot be relied upon to produce true successful outcomes. For the future, we should begin to work on potent oral antivirals against all major classes of potential pathogens, with the goal of having drugs ready for Phase 2/3 efficacy trials when the next threat emerges.

Our collaboration identified key drug repurposing opportunities for COVID-19 treatment and prevention and highlighted the importance of standardized testing of preclinical data such as PK exposure when interpreting the emerging candidacy of drugs for COVID-19 treatment and prevention. Overall, we believe that following the standardized process to assess repurposed drugs described in Figure 1, we could improve translation of selected repurposed compounds to clinical trials. It is interesting to note that the current WHO guidelines and recommendations for the use of given drugs [79] reflect our observations, in that nirmatrelvir and molnupiravir are recommended for COVID-19 treatment, whereas ivermectin, hydroxychloroquine, and lopinavir/ritonavir are not recommended. Further drugs will be added to the list once robust clinical data are available.

We advocate a multi-pronged approach to drug repurposing for COVID-19 treatment, using the preclinical tests presented in the current study, including carrying out the same assay in different laboratories across different cell lines. This pragmatic approach is intended to build confidence in the validity of preclinical data. No one model fits all, but several strategies are required depending on the MoA of the drug.

Identification of efficacious drugs against COVID-19 will reduce viral load in patients, reduce hospitalizations, and ultimately relieve healthcare systems. Sobering data from the WHO point to the impact of the pandemic and to the need for countries to invest in more resilient health systems that can sustain essential health services during crises, including stronger health information systems [5]. The lessons learned will be critical in improving the response and control of future pandemics.

## Figures and Tables

**Figure 1 microorganisms-10-01639-f001:**
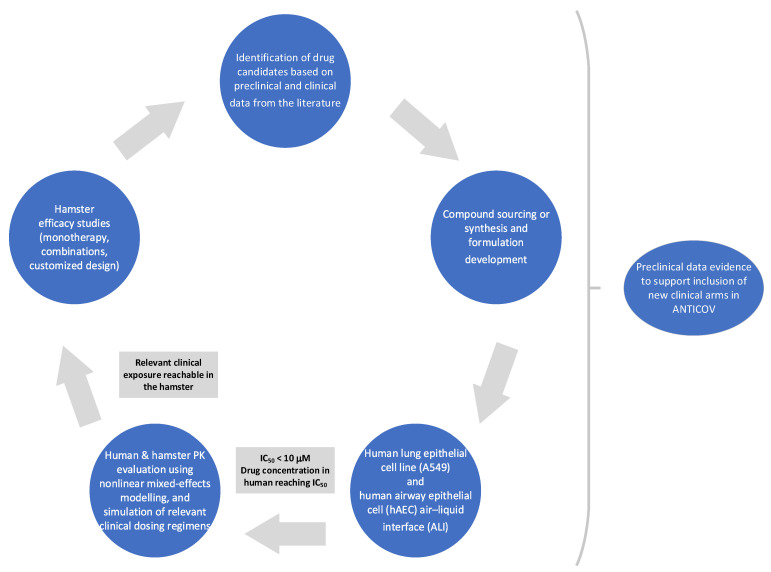
Ideal process for generating new preclinical data of repurposed drugs against SARS-CoV-2 to build a rationale for a clinical evaluation.

**Figure 2 microorganisms-10-01639-f002:**
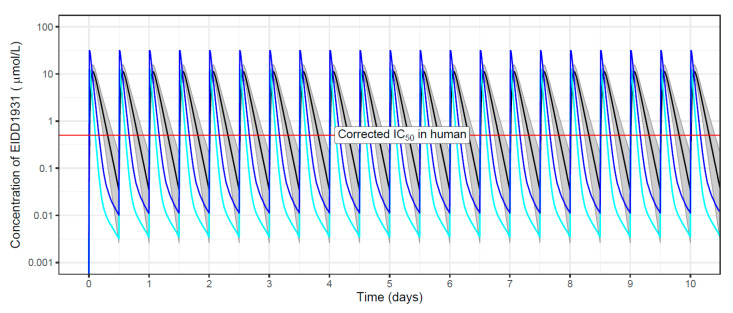
Simulated plasma concentration–time profiles for the molnupiravir metabolite EIDD-1931 in humans (70 kg) and hamsters (120 g), based on the final population pharmacokinetic models (for details, see Appendix A). The black line (bounded by a grey shade showing the 90% confidence interval) represents the median simulated concentration-vs.-time profile in humans at a clinically relevant dose (800 mg molnupiravir, twice daily, for 10 days). The cyan line represents the median profile in hamsters, following administration of 50 mg/kg molnupiravir, twice daily for 10 days, corresponding to the approximate dose in hamsters matching C_max_ in humans. The dark blue line represents the median profile in hamsters following administration of 150 mg/kg molnupiravir, twice daily for 10 days, corresponding to the approximate dose in hamsters matching AUC_total_ in humans; 150 mg/kg twice daily was also the efficacious dose in the hamster infection model of SARS-CoV-2. The horizontal red line denotes the IC_50_ (A549-ACE2TMPRSS2 cells), corrected for plasma protein binding in hamsters and humans (Appendix A).

**Figure 3 microorganisms-10-01639-f003:**
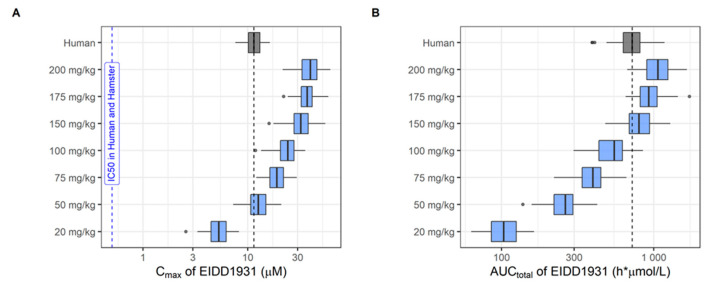
Simulated pharmacokinetic parameters of molnupiravir in humans and hamsters: (**A**) C_max_; (**B**) AUC_total_. Predicted pharmacokinetic parameters, i.e., C_max_ and AUC_total_, in humans receiving 800 mg of molnupiravir twice daily for 10 days (grey box) were compared with pharmacokinetic parameters of hamsters receiving 20 to 200 mg/kg of molnupiravir twice daily (blue boxes) for same number of days. Boxes and whiskers represent the median with inter-quantile range and the 95% prediction intervals, respectively. The blue vertical line denotes the IC_50_ (A549-ACE2TMPRSS2 cells) corrected for plasma protein binding in hamsters and humans.

**Table 1 microorganisms-10-01639-t001:** Overview of prioritized repurposed drugs and generated preclinical data–Single agents.

Repurposed Drug/Experimental Compound	Mechanism of Action; Target	Activity In Vitro; Potency EC_50_ or IC_50_ < 10 µM			Activity Ex Vivo Human Airway Epithelia (HAE)	Exposure in Human at Clinically Relevant Dose and Matching Doses in Hamster ^a^			Activity and Exposure in Hamster Infection Model of SARS-CoV-2			
		Vero Cells	Calu-3 Cells	A549 Cells		C_max_ Human Above IC_50_ at Clinically Relevant Dose	C_max_ Hamster Above IC_50_ for Dose Matching C_max_ in Human	C_max_ Hamster Above IC_50_ for Dose Matching AUC in Human	Activity in SARS-CoV-2 Hamster Model	Dose(s) per Occasion, Frequency, Duration	C_max_ Hamster Above IC_50_ ^b^	C_min_ Hamster Above IC_50_ for >24 h ^b^
**Direct Acting Antivirals (DAA)**												
Atazanavir (ritonavir-boosted)	HIV protease inhibitors	No	No	No	No	No	No (24 mg/kg)	No (72 mg/kg)	No	48 mg/kg (16 mg/kg ritonavir), BID, 3–4 days	No (48/16 ritonavir mg/kg, BID)	No (48/16 mg/kg ritonavir, BID)
Bemnifosbuvir (AT-527/AT-511) (experimental cpd) ^c^	RdRp (guanosine nucleotide analogue)	No	No	No	No	Yes	ND		No ^d^	150–250 mg/kg, BID, 3 days	No (250 mg/kg, BID)	No (250 mg/kg, BID)
Daclatasvir	HCV NS5A inhibitor (polymerase inhibitor)	Yes ^Pgp; T^	No	No ^T^	No	No	No (10 mg/kg)	No (15 mg/kg)	No	25 mg/kg, BID, 3 days	No (25 mg/kg, BID)	No (25 mg/kg, BID)
Favipiravir	RNA-dependent RNA polymerase (RdRp),	No	No	No	No	No	Yes (25 mg/kg)	Yes (25 mg/kg)	Yes	300 mg/kg, BID, 4 days; 462.5 mg/kg, BID, 3 days	Yes (300 mg/kg, BID)	No (300 mg/kg, BID)
Molnupiravir and metabolite EIDD-1931, experimental cpd/Merck)	RdRp (hydroxy-cytidine nucleotide analogue)	Yes ^Pgp^	Yes	Yes	Yes (10 µM)	Yes	Yes (50 mg/kg)	Yes (150 mg/kg)	Yes	75–200 mg/kg, BID, 3–4 days	Yes (150 mg/kg BID)	No (150 mg/kg BID)
Nelfinavir	HIV protease inhibitors	Yes ^T^	Yes (borderline, 13 µM)	Yes ^T^	Yes	No	No (70 mg/kg)	No (100 mg/kg)	No ^e^	100 mg/kg, QD, cepharanthine boosted (50 mg/kg, BID)	No (100 mg/kg)	No (100 mg/kg)
Nirmatrelvir (experimental cpd/Pfizer) [35]	Mpro inhibitor	Yes	Yes	Yes	Yes (1 µM)	ND			Yes	125–250 mg/kg, BID, 4 days	Yes (125 mg/kg, BID)	Yes (125 mg/kg, BID)
Sofosbuvir	HCV NS5B inhibitor (polymerase inhibitor)	No	No	No	No	No	No (>200 mg/kg)	No (>200 mg/kg)	No	100 mg/kg, QD, 3 days	No (100 mg/kg)	No (100 mg/kg)
**Other Mechanisms of Action—Indirect Acting Antivirals**												
Ambroxol	Mucolytic/prevent virus to bind to ACE-2 receptor	No	No	No	No	No	No (30 mg/kg)	No (50 mg/kg)	No	50 mg/kg, BID, 3 days	No (50 mg/kg BID)	No (50 mg/kg BID)
Amodiaquine ^f^	Anti-malarial	Yes (~10 µM)	No	No	No	No	No (<5 mg/kg)	No (<5 mg/kg)	No (parent)	50–100 mg/kg, QD, 4–5 days	No (100 mg/kg)	No (100 mg/kg)
Cepharanthine	Block virus entry/ ACE2 binding	Yes ^T^	No	No	No	No	No (1 mg/kg)	No (1 mg/kg)	No	100 mg/kg, QD, 4 days	No (100 mg/kg)	No (100 mg/kg)
Camostat mesylate	TMPRSS2 inhibitor	No	Yes	No	Yes (10 µM)	ND			No	200 mg/kg, BID, 4 days	ND	
Clofazimine	TB inhibitor/ accumulate in lungs	Yes ^T^	Yes	Yes ^T^	No	No	No (1 mg/kg)	No (1 mg/kg)	No	25 mg/kg, QD, 4 days	Yes (25 mg/kg)	No (25 mg/kg)
Colchicine	Anti-inflammatory	Yes ^T^	Yes ^T^	Yes ^T^	No	ND			NT			
Fluoxetine	SSRI (selective serotonin reuptake inhibitor	Yes ^T^	No	No	No	No	No (10 mg/kg)	No (10 mg/kg)	No	10–100 mg/kg, QD, 4 days	No (100 mg/kg)	No (100 mg/kg)
Fluvoxamine maleate	SSRI (selective serotonin reuptake inhibitor	No	No	No	No	No	No (12 mg/kg)	No (20 mg/kg)	No	100 mg/kg, QD, 3 days; 200 mg, BID, 4 days	No ^g^ (100 mg/kg)	No ^g^ (100 mg/kg)
Ivermectin (oral)	Anti-parasitic drug, Anti-inflammatory	Yes ^T^	Yes	Yes ^T^	Yes ^T^	No	No (0.1 mg/kg)	No (0.1 mg/kg)	NT			
Ivermectin (s.c.)						No	No (0.1 mg/kg)	No (0.1 mg/kg)	No	0.4 mg/kg, QD, 1 day or 4 days	No (0.4 mg/kg)	No (0.4 mg/kg)
Mefloquine	Anti-malarial	Yes ^T^	No	No	No	ND			NT			
Nitazoxanide (and metabolite tizoxanide)	Antiprotozoal agent (*Giardia*, *Cryptosporidium* infections)	Yes	Yes ^Pgp^	No	Yes	No	No (25 mg/kg)	No (150 mg/kg)	No	250 mg/kg, BID, 3–4 days	No (250 mg/kg BID)	No (250 mg/kg BID)
Pentoxyfilline	Vasodilatator; anti-inflammatory	No	No	No	No	ND			NT			
Probenecid	Anti-gout?	No	No	No	No	ND			NT			
Proxalutamide	Androgen receptor antagonist, Anti-inflammatory	No ^T^	No	No ^T^	No	ND			NT			

^a^ Plasma exposure in humans at clinically relevant dose (once or twice daily dosing, according to the product information) and hamsters (twice daily dosing) were simulated using population pharmacokinetic models listed in the Appendix A. Simulations were performed for a total treatment duration of 10 days. Dosing information in humans is provided in Appendix A, along with details for derivation of doses in hamsters matching C_max_ and AUC_total_ in humans (doses provided in brackets). IC_50_ refers to in vitro activity in A549-ACE2TMPRSS2 cells (if not otherwise indicated) and following correction for protein binding in medium and human/ hamster plasma (Appendix A). ^b^ Dose in hamsters used for simulations of C_max_ refer to the maximum dose tested in the hamster infection model of SARS-CoV-2 (if no activity was observed) or the minimum efficacious dose. Twice daily dosing was assumed for simulations. ^c^ AT-273 is AT-527 metabolite measured in plasma and used as a surrogate for AT-527 plasma concentration; human PK parameters based on literature data. ^d^ no activity in hamster, except for significant reduction in RNA Yields plasma log_10_ [copies/mL]; ^e^ no effect on virus load, but effect on lung inflammation and “disease” outcome; ^f^ IC_50_ data for Des-Ethyl-Amodiaquine (=amodiaquine metabolite) not available; ^Pgp^ in vitro assay was performed in the presence of a P-glycoprotein inhibitor; ^T^ toxicity observed; highlighted in blue: exposure (C_max_ or C_min_) above IC_50_ in hamsters or humans and/or activity in the hamster infection model of SARS-CoV-2; AUC, area under the plasma concentration-time curve; BID, twice daily; C_max_, peak plasma concentration; cpd, compound; DHE, Des-Ethyl-Amodiaquine (=amodiaquine metabolite); EC_50_ and IC_50_, effective or inhibitory concentration leading to half-maximum activity (when not generated experimentally, IC_50_ were derived from literature (see also Appendix A); HCV, hepatitis C virus; HIV, human immunodeficiency virus; NA, not applicable; ND, not determined; NT, not tested; PD, pharmacodynamic; QD, once daily; r, ritonavir; RdRp, RNA-dependent RNA polymerase; s.c., subcutaneous; TB, tuberculosis; ^g^ 200 mg/kg dose was toxic.

**Table 2 microorganisms-10-01639-t002:** Overview of prioritized repurposed drugs and generated preclinical data- Combination regimen.

	Mode of Action	Dose (mg/kg/day)	Efficacy In Vivo SARS-CoV-2 Hamster Model	Comments
Atazanavir /ritonavir (ATZ/r) /Nitazoxanide	DAA/IAA combination	96/32/500 mg/kg/day	No	
FAV/ATZ/r	DAAs combination	600/96/32 mg/kg/day	Yes	No additive or synergistic activity as compared with FAV alone
FAV/Nitazoxanide	DAA/IAA combination	600/500 mg/kg/day	Yes	No additive or synergistic activity as compared with FAV alone
Sofosbuvir/daclatasvir	DAAs combination	100/100 mg/kg/day	No	
Nelfinavir/Cepharantine	DAA/IAA combination	100/100 mg/kg/day	No	
Ivermectin/Amodiaquine	DAA (considered)/IAA combination	0.4/50 mg/kg/day	No	
Molnupiravir/Clofazimine	DAA/IAA combination	150/25 mg/kg/day	Yes	No additive or synergistic activity as compared with Molnupiravir alone
Molnupiravir/Nirmatrelvir	DAA/DAA combination	150/250 mg/kg/day	Yes	No additive or synergistic activity as compared with nirmatrelvir alone Additive effect as compared with Molnupiravir alone

ATZ, atazanavir; DAA, direct acting antiviral; FAV, favipiravir; IAA, indirect acting antiviral; r, ritonavir.

**Table 3 microorganisms-10-01639-t003:** Target exposure vs. activity in the hamster infection model of SARS-CoV-2.

C_max_ in Hamster Above Corrected IC_50_	Activity in Hamster Infection Model of SARS-CoV-2		Total
	Yes	No	
**Yes**	TRUE POSITIVE (TP) *n* = 3 (favipiravir, molnupiravir, nirmatrelvir)	FALSE POSITIVE (FP) *n* = 1 (clofazimine)	4
**No**	FALSE NEGATIVE (FN) *n* = 0	TRUE NEGATIVE (TN) *n* = 12 (atazanavir, bemnifosbuvir, daclatasvir, nelfinavir *, sofosbuvir, ambroxol, amodiaquine, cepharanthine, fluoxetine, fluvoxamine, ivermectin, nitazoxanide	12
Total	3	13	16

* no effect on viral load was detected, but a diminution of lung inflammation and “disease” outcome was nevertheless observed; sensitivity: 100% (TP/TP + FN); specificity: 92% (TN/FP + TN).

## Data Availability

Not applicable.

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
