# Peer review of "Need for a Standardized Translational Drug Development Platform: Lessons Learned from the Repurposing of Drugs for COVID-19"

_microorganisms, 2022, doi:10.3390/microorganisms10081639_

Round 1

Reviewer 1 Report

REVIEW OF ASSMUS, et al.

Assmus, et al examined drug repurposing for SARS-CoV-2 using a hamster model and cell based assays.  They showed that molupiravir, favipiravir, clofazimine nirmatrelvir, and daclatasvir were effective. They also simulated PK data using R.  Interestingly, the Indirect acting antivirals showed no effect in the hamster mode.  The paper is interesting to read, and the methods appear to be up to date.  I only have minor revisions.

MINOR

11.       Introduction and Line 164 -  I am surprised that you did not mention the anticoagulation therapeutics. These were given to about 30% of the COVID inpatients in the US. Blood clotting was to some degree a surprise to the physicians even though it was predicted by bioinformatics (PMID 34019767).   When I think of drug repurposing  I don’t expect a silver bullet, but rather a cocktail or a regimen. I think anticoagulants here were a type of drug repurposing, since they aren’t typically used for respiratory illnesses.  Is there some way you can mention anti-coagulants?  I suspect that the mortality would have been higher had these drugs not been used.  I think it also underscores the fact that SARS was not a typical respiratory infection, it was atypical and most clinicians hadn’t seen a SARS patient prior to the pandemic.

22.       Line 117, Intro – Please include the dates when vaccines were available to physicians and at risk patients.  In the US healthcare workers received the vaccine in Dec 2021, and the general pop received it in Feb-Mar 2021.

33.       Line 132, Intro; Line 763 – I think its important to note that we had global public health information from Johns Hopkins School of Public Health.  “Flattening the curve” was largely in ref to the Hopkins dashboard graph on the bottom right hand corner, maybe even show a picture of it?  I think if I were to read this paper 40 years from now, I wouldn’t know what you had access to (e.g. GISAID).

44.       Tables – please reformat your tables with a smaller font, these are too hard to read.

55.       Why is remdesivir not on your list?  You should add some type of reasoning in the results/discussion.

66.       Line 284 – can you briefly list the pathology that occurs in hamsters?

77.       Line 378 – what criteria did you use for down-select?

88.       Line 442 – add the Mech of action of these drugs in a sentence or 2.

99.       Line 587 – ‘rapidly evolved’ I would strongly disagree with this statement.  The enzymes of these (+)ssRNA Group IV viruses don’t tend to mutate quickly.  The fidelity of the polymerase is very different. They are definitely different from Group VI (e.g. HIV).  The spike protein mutates, but the signs/symptoms of SARS-CoV-2 are still the same as in 2020. Mutation of the structural proteins of a virus may affect vaccine efficacy and tropism, but probably won’t affect antiviral efficacy.  You might want to delete 587-591. I don’t believe this supported by clinical or sequence data since there were significantly more omicron cases than alpha and beta.  Changes in the UTR’s tend to attenuate these viruses, this is something we’ve learned from live virus strains for Group IV’s.

110.   Line 612 – may have an impact

111.   Table 2 – add horizonal lines or a box

Author Response

Assmus, et al examined drug repurposing for SARS-CoV-2 using a hamster model and cell based assays. They showed that molupiravir, favipiravir, clofazimine nirmatrelvir, and daclatasvir were effective. They also simulated PK data using R. Interestingly, the Indirect acting antivirals showed no effect in the hamster mode. The paper is interesting to read, and the methods appear to be up to date. I only have minor revisions.

We thank Referee 1 for these very positive comments.

MINOR

Introduction and Line 164 - I am surprised that you did not mention the anticoagulation therapeutics. These were given to about 30% of the COVID inpatients in the US. Blood clotting was to some degree a surprise to the physicians even though it was predicted by bioinformatics (PMID 34019767).   When I think of drug repurposing  I don’t expect a silver bullet, but rather a cocktail or a regimen. I think anticoagulants here were a type of drug repurposing, since they aren’t typically used for respiratory illnesses.  Is there some way you can mention anti-coagulants?  I suspect that the mortality would have been higher had these drugs not been used.  I think it also underscores the fact that SARS was not a typical respiratory infection, it was atypical and most clinicians hadn’t seen a SARS patient prior to the pandemic.

Antigoagulation drugs have indeed been used in COVID-19 patients. As such these drugs were part of repurposed drugs showing an impact in the pandemic; thank you for the reminder. We have added this fact (line 169-line 174 in the revised manuscript) also highlighting the limitation of the use of such drugs to a specific patient population (as depicted in rference 23, the current cited NIH guidelines on the topic).

Line 117, Intro – Please include the dates when vaccines were available to physicians and at risk patients.  In the US healthcare workers received the vaccine in Dec 2021, and the general pop received it in Feb-Mar 2021.

Thank you for the suggestion. This important milestone in the COVID-19 pandemic has been added (line 117-line 118 in the revised manuscript).

Line 132, Intro; Line 763 – I think its important to note that we had global public health information from Johns Hopkins School of Public Health.  “Flattening the curve” was largely in ref to the Hopkins dashboard graph on the bottom right hand corner, maybe even show a picture of it?  I think if I were to read this paper 40 years from now, I wouldn’t know what you had access to (e.g. GISAID).

We agree that it is important to mention the global public health information that was available on COVID-19, in particular those available through the Coronavirus Resource Center of John Hopkins university of medicine, that indeed led to this “flattening of the curve” wording. This has been added together with the corresponding reference 4 (line 134-line 136 in the revised manuscript)

Tables – please reformat your tables with a smaller font, these are too hard to read.

The Tables were originally formatted according to the journal’s instructions for authors. The format of the Tables appears to have changed during the upload of the manuscript. We have reformatted them accordingly and hope these are now easier to read. While it makes the document easier to read, we regret that these formatting changes were not tracked for Table 1 and 2, but confirm that only the formatting was changed.

Why is remdesivir not on your list?  You should add some type of reasoning in the results/discussion.

We have added a brief rationale in the Results section (line 422-line 426 in the revised manuscript) to explain the reason for not including remdesivir in the analysis and further details in the Supplementary Material, Section 1.4, supported by several references (line 124-line 137 in the revised supplement). Briefly, the fact that it had to be administered by infusion in human was not considered adequate for inclusion in ANTICOV; moreover, it was not deemed a suitable control for Syrian hamster experiments (difficult to administer and lack of efficacy in standard rodent animal models of SARS-CoV2 infection due to poor plasma stability in this species).

Line 284 – can you briefly list the pathology that occurs in hamsters?

We have briefly described the features associated with SARS-CoV-2 infection in hamsters (line 299-line 304 in the revised manuscript) and refered to further details on the relevance and pathology description in hamster following SARS-CoV-2 infection in the Supplementary Materials, Section 1.4 (line 115-123 in the revised supplement supported by additional references).

Line 378 – what criteria did you use for down-select?

Criteria used for down selection from 88 down to 25 compounds for further preclinical assessment are explained in paragraph “2.1. Identification and selection of drug repurposing candidates for preclinical studies” (line 221 onwards in the revised manuscript). We added “down-selecting” in Line 243 and “using criteria listed in 2.1 (Identification and selection of drug repurposing candidates for preclinical studies)” in line 419-line 420 to add clarification.

Line 442 – add the Mech of action of these drugs in a sentence or 2.

We have added two sentences mentioning the different mechanism of action (MoA) of the drugs tested (line 430-line 435). We reiterated the MoA of the DAAs in line 512-line 515. MoAs for the drug tested are also depicted in Table 1A (second column).

Line 587 – ‘rapidly evolved’ I would strongly disagree with this statement.  The enzymes of these (+)ssRNA Group IV viruses don’t tend to mutate quickly.  The fidelity of the polymerase is very different. They are definitely different from Group VI (e.g. HIV).  The spike protein mutates, but the signs/symptoms of SARS-CoV-2 are still the same as in 2020. Mutation of the structural proteins of a virus may affect vaccine efficacy and tropism, but probably won’t affect antiviral efficacy.  You might want to delete 587-591. I don’t believe this supported by clinical or sequence data since there were significantly more omicron cases than alpha and beta.  Changes in the UTR’s tend to attenuate these viruses, this is something we’ve learned from live virus strains for Group IV’s.

We have deleted this text accordingly, as suggested.

Line 612 – may have an impact

“have an” has been added in the sentence (line 698 in the revised manuscript).

Table 2 – add horizonal lines or a box

Thank you. We have added an horizontal line as suggested.

Reviewer 2 Report

In the manuscript (Need for a standardized translational drug development plat- 2 form: lessons learned from the repurposing of drugs for COVID-19), the authors stated the challenges involved in rapidly identifying antiviral treatments and the need for an improved and standardized drug development process. The authors stated that a unified standardized strategy is necessary for selecting, testing, and validating candidate drugs. The authors used a number of methods and identified key drug repurposing opportunities for COVID-19 treatment and prevention and highlighted the importance of standardized testing of preclinical data such as PK exposure when interpreting the emerging candidacy of drugs for COVID-19 treatment and prevention. The paper is well designed and organized.

-What are the limitations of this study? Please mention in discussion section.

-Insert some more references in the methods section especially in "supplementary methods" to support your methods.

-The results are based on the presented data and the quality of figures is okay.

-In figure 3 legends, mention what is shown in A and B?

Author Response

In the manuscript (Need for a standardized translational drug development plat- 2 form: lessons learned from the repurposing of drugs for COVID-19), the authors stated the challenges involved in rapidly identifying antiviral treatments and the need for an improved and standardized drug development process. The authors stated that a unified standardized strategy is necessary for selecting, testing, and validating candidate drugs. The authors used a number of methods and identified key drug repurposing opportunities for COVID-19 treatment and prevention and highlighted the importance of standardized testing of preclinical data such as PK exposure when interpreting the emerging candidacy of drugs for COVID-19 treatment and prevention. The paper is well designed and organized.

We thank Referee 2 for these very positive comments.

What are the limitations of this study? Please mention in discussion section.

We have added two sentences to further strengthen the main limitations of the study (line 865-line 870 of the revised manuscript), namely the identification of potential repurposed drugs of interest having another MoA than solely antiviral activity and achieving clinically relevant exposure in animal models (need for optimal formulation among others and different metabolism in animal species) to increase confidence in the potential of these drugs in human.

Insert some more references in the methods section especially in "supplementary methods" to support your methods.

We have reformatted the supplementary material document, bringing back in the methodology section Tables corresponding to the experimental conditions for in vivo testing, PK assessment in hamsters and bioanalytical details. We have added more details including references related to the relevance of the models and experimental controls used among others.

The results are based on the presented data and the quality of figures is okay.

We thank Referee 2 for these very positive comments.

In figure 3 legends, mention what is shown in A and B?

This had indeed been forgotten and has been added in Figure 3 legend.
